# 3DNA: A Tool for Sculpting Brick-Based DNA Nanostructures †

**Shikhar Kumar Gupta** [1,2], **Foram Joshi** [1,2], **Amay Agrawal** [1,2], **Sourav Deb** [1], **Martin Sajfutdinow** [2], **Dixita Limbachiya** [1], **David M. Smith** [1,2,3,*] and **Manish K. Gupta** [1,*]

1    Laboratory of Natural Information Processing, Dhirubhai Ambani Institute of Information and Communication Technology, Gandhinagar 382007, India
2    DNA Nanodevices, Fraunhofer Institute for Cell Therapy and Immunology, 04103 Leipzig, Germany
3    Peter Debye Institute for Soft Matter Physics, University of Leipzig, 04103 Leipzig, Germany
*    Correspondence: david.smith@izi.fraunhofer.de (D.M.S.); mankg@guptalab.org (M.K.G.);
     Tel.: +49-341-35536-9311 (D.M.S.); +91-79-68261549 (M.K.G.)
†    A preliminary version of this article appeared as a poster paper at the 11th Annual Conference on Foundations of Nanoscience: Self-Assembled Architectures and Devices 2014 (FNANO14), and the corresponding preprint can be found in https://arxiv.org/pdf/1405.4118.pdf.

**Abstract:** To assist in the speed and accuracy of designing brick-based DNA nanostructures, we introduce a lightweight software suite 3DNA that can be used to generate complex structures. Currently, implementation of this fabrication strategy involves working with generalized, typically commercial CAD software, ad-hoc sequence-generating scripts, and visualization software, which must often be integrated together with an experimental lab setup for handling the hundreds or thousands of constituent DNA sequences. 3DNA encapsulates the solutions to these challenges in one package by providing a customized, easy-to-use molecular canvas and back-end functionality to assist in both visualization and sequence design. The primary motivation behind this software is enabling broader use of the brick-based method for constructing rigid, 3D DNA-based nanostructures, first introduced in 2012. 3DNA is developed to provide a streamlined, real-time workflow for designing and implementing this type of 3D nanostructure by integrating different visualization and design modules. Due to its cross-platform nature, it can be used on the most popular desktop environments, i.e., Windows, Mac OS X, and various flavors of Linux. 3DNA utilizes toolbar-based navigation to create a user-friendly GUI and includes a customized feature to analyze the constituent DNA sequences. Finally, the oligonucleotide sequences themselves can either be created on the fly by a random sequence generator, or selected from a pre-existing set of sequences making up a larger molecular canvas.

**Keywords:** CAD software; DNA origami; DNA bricks; DNA nanostructures; GC-content; LEGO®; DNA structural stability

## 1. Introduction

In modern nanofabrication, the requirements of both precision and scalability create the demand for new technologies that simultaneously meet these needs. Of the most promising approaches for nanoscale fabrication, working with biological components such as DNA, bacteria, or viruses has become an exponentially attractive topic among researchers in recent decades. In particular, the construction of nanoscale devices using the self-assembly of DNA has led to an explosion of applications in fields as diverse as nano-plasmonics [1], fabrication of nanoscale electronic components [2], and various areas of biotechnology [3?,4]. These methods significantly impact academic research and are slowly making inroads into real-world, commercial applications. Based on the principle that DNA strands hybridize with corresponding strands containing the complementary sequences of bases, sets of DNA oligonucleotides can be programmed in a specific manner that enforces the assembled structure to form a particular shape. This can be accomplished

using a variety of techniques that have been established over the last few decades since the concept was first suggested by Nadrian Seeman in 1982 [6].

In 2006, Paul Rothemund revolutionized the approach of making complex, nanoscale shapes using DNA by introducing the concept of DNA Origami [7], which was subsequently expanded into three dimensions in 2009 by Douglas et al. [8]. The core idea is that a long scaffold DNA sequence (typically several thousand bases long) can be folded into specific shapes using short synthetic DNA strands, containing sections that are complementary to the respective components of the scaffold. While software tools were quickly developed to aid in the design of 3D DNA origami structures [9], the technique was still limited by the discrete size of structures enforced by the scaffolded approach, as well as certain limitations for topological complexity due to the primary method of folding a large DNA strand into the desired shape.

Several years later in 2012, Wei et al. introduced the single-strand DNA tiles approach to construct two-dimensional DNA nano-objects [10] through the assembly of short oligonucleotide "tiles", eliminating the need for an underlying scaffold strand. Simultaneously, Ke et al. [11] expanded upon this scaffold-less approach, proposing a new direction for generating complex DNA-based objects based upon the self-assembly of several hundred single-stranded DNA "bricks" on a 3D molecular canvas of individual voxels. These short DNA oligonucleotide sequences (referred to as DNA bricks) are designed to self-assemble into predefined shapes in a one-step, thermal annealing reaction. The underlying molecular architecture of how individual bricks are interconnected to their neighbors and how these motifs can be used to construct complex objects was shown to be analogous to the assembly of LEGO® bricks [12]. Unlike individual DNA origami structures, the shapes made using DNA bricks are not limited by the size of the DNA scaffold; discrete structures having been built out of 10,000 or more individual components [13], and micron-sized surfaces have been constructed through the crystalline repeat of smaller motifs [14]. Hence, this method provides more flexibility in making shapes by utilizing a modular approach to DNA self-assembly and thus holds potential promise for industrial nanofabrication of large substrates.

Despite these advantages, particularly in creating extended nanoscale patterning of molecules and other components over large surface areas, the significant complexity of designing large DNA brick structures from several hundred or even several thousand components has limited its more comprehensive application. Perhaps the most frequent use of this assembly method has been to inspire a wealth of theoretical and modeling-based investigations into the self-assembly of multi-component structures. More than simply understanding the self-assembly process and impact of individual molecular aspects on the overall structural features [15–19], several studies have suggested strategies to optimize factors such as sequence design or assembly protocols [20–24] as well as predict the in situ properties such as conductivity of assembled structures [25]. Experimental studies tested and elaborated upon some of these in silico findings, specifically examining the impact [26] and strategical exploitation [27] of brick nucleation for optimizing assembly yields. Brick-based motifs and specific 3D structures have been used to either develop or demonstrate new strategies for assembling particular types of structures such as tubes and ribbons [28,29], complex polygons [30], compact cubes [31] and large super-structures [32], or as a convenient testing ground to demonstrate strategies to enhance biocompatibility [33] or library-based production of oligonucleotide sets [34]. Nevertheless, the exploitation of this method in studies that point towards its enormous potential in the area of nano-fabrication has been few and far between. Notably, discrete DNA brick structures have been used as precise breadboards for the placement of exciton-based molecular circuits [35], and large-scale arrays of semiconducting carbon nanotubes have been generated through their topological confinement in so-called "nanotrenches" in extended DNA brick lattices [36]. With the recent development of strategies to minimize defects in extended crystalline arrays of DNA brick motifs [37], molecular tools for the broad usage of this method are in place.

However, dedicated software tools for designing structures are still necessary to bring the method forward.

DNA nanostructures can be designed and constructed by different approaches for a broad range of applications in research. However, experimentally implementing such a solution involves effective exploitation of a specific design strategy and a high degree of expertise in experimental practice. Hence, a substantial advantage for various applications can be derived from dedicated software to design artificially programmable DNA structures according to the different methods. Several Computer-Aided Design (CAD) software packages have been developed through the last decades for the different DNA-based assembly approaches [38]. While the most well-known CAD software for designing scaffolded DNA origami is Cadnano, initially developed in 2009 [9], many other design tools such as Tiamat [39], vHelix [40], and the newly introduced ENSnano [41] have been implemented in recent years. Moreover, a class of closely related software, DAEDALUS [42], PERDIX [43], TALOS [44], and METIS [44], have been introduced for the design of scaffold-based DNA wireframe structures. Departing from the DNA origami method, the program DNA Pen [45] can be seen as a software package targeted exclusively for designing 2D complex nanostructures based on the 2D DNA tile method. The latest addition to this approach can be found as the software DNAxiS [?] to design oblique-shaped axially symmetric closed DNA nanostructure. DNAxiS is identified as the only multilayer design scheme that blends probabilistic algorithmic approach and numerical approximation measures for DNA scaffold crossover placement and overall orientation.

In addition to the above-mentioned software packages for designing various nucleotide architectures, numerous tools enabling their in-silico modeling and visualization have also been introduced over the last two decades. Early programs such as Mfold for the analysis of RNA and DNA folding [47], or the similarly named 3DNA (later w3DNA) for reconstructing 3D DNA motifs from protein database (PDB) files [48,49] were introduced shortly after the turn of the millennium. More recently, these have been supplemented with a large variety of web-based and downloadable programs for investigations into complex 3D DNA nanostructures at varying degrees of detail [38]. While not CAD programs themselves, they do form an important complement to design programs as they can give a preliminary indication of structural features, particularly in the equilibrated, post-assembly state.

However, no such dedicated software solution exists for rapidly prototyping DNA-based nanostructures according to the DNA brick method, beginning from basic topological design to the output of (and control over) corresponding sets of oligonucleotide sequences. In the original introduction of the method in 2012 [11], the authors used a combination of commercial CAD software with self-programmed scripts to enable their flexible assembly method. Existing software for DNA origami, specifically Cadnano, can be used to develop brick-like architectures with roundabout design tricks. However, the lack of an editable, 3D molecular canvas limits the creation of some complex architectures containing cavities or the straightforward design of extended surfaces.

Thus motivated, we introduce a software suite, 3DNA, to model, edit, and visualize complex structures of single-stranded DNA Bricks to overcome the technical difficulties in a lab setup. From the application point of view, the critical advantage is achieving specific 3D DNA shapes with greater accuracy in less time. This open-source software's productivity lies in reducing the error-proneness in designing DNA sequences to generate the specific shapes, while concurrently reducing the amount of time required to progress from a design concept to the set of constituent DNA sequences. The software also includes modules for generating DNA sequences, which can later be used to envisage the designed shape in the laboratory. While a rigorous experimental demonstration of a variety of different designs created using this software should still be carried out, the numerous beneficial features of 3DNA validate the theoretical underpinnings, demonstrating its relevance and potential to solve practical problems, ultimately enhancing its value and impact.

It is also noteworthy that although our proposed software platform and the previous software package [48] developed by Lu et al. share similar names and involve working with 3D DNA structures at the nanoscale, they serve different purposes. While the primary focus of the 3DNA software package developed by Lu et al. was the visualization of DNA nanostructures, our proposed 3DNA platform contributes to the design and stabilization of the DNA nanostructures in 3D format.

## 2. Methods and Implementation

3DNA executes a two-stage algorithm [50] in the intermediate process to generate the desired nanostructure. In Stage I, domain sequences are generated by designing the 32-nt full and 16-nt half brick structures involving four 8-nt domains and two 8-nt domains, respectively.

Figure 1 represents a full brick and a half brick. Notably, this design algorithm considers some universally fundamental biological constraints on the input set of randomly assigned DNA sequences, in particular, avoidance of runs of length more than 4 and Hamming distances of higher than 6 per domain. According to practical considerations for oligonucleotide synthesis and correct thermal assembly, 40–60% GC-content for the set of sequences is also considered for sequence generation.

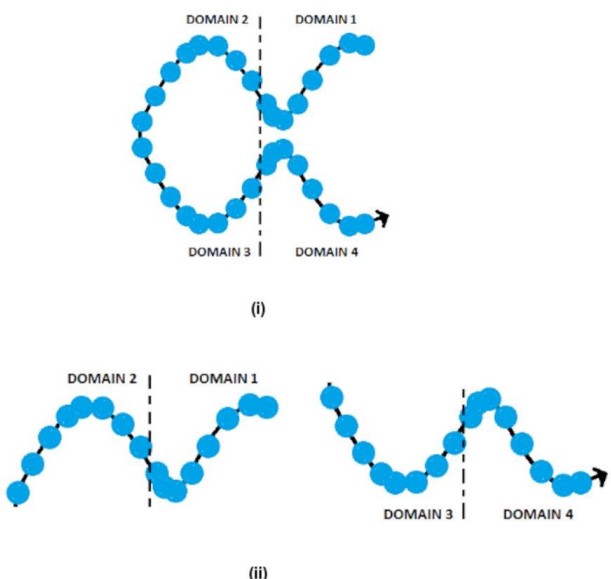

**Figure 1.** The single-stranded DNA structure of (**i**) full bricks having four 8-nt domains, and (**ii**) half bricks having two 8-nt domains [50].

The subsequent step associates brick sequence-generation from molecular pixels to generate the relevant sculpture in Stage II.

The brick orientation is accounted for by visualizing each brick with four connective domains analogous to simple LEGO® 1 × 2 bricks [12]. Each brick is attached to the four complementary adjacent neighbors in a LEGO®-like manner. The directional orientation in each of these junctions is set in the perpendicular pattern since each brick follows a horizontal or vertical orientation in the three-dimensional space. A pictorial illustration is given in Figure 2.

In parallel, the Graphic User Interface (GUI) of 3DNA is designed to enable an intuitive and user-friendly environment for creating structures on the 3D modular canvas, even for users with minimal in-depth knowledge of DNA bricks. It consists of three main modules: the 3D molecular canvas, modeling, and DNA sequence generation. The interface (Figure 3) consists of basic information and elementary options for the users for different experiences.

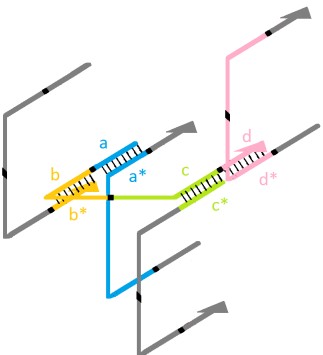

**Figure 2.** The full brick structure formation utilizing complementary DNA strands (indicated in identical colors) present in the respective domains [50].

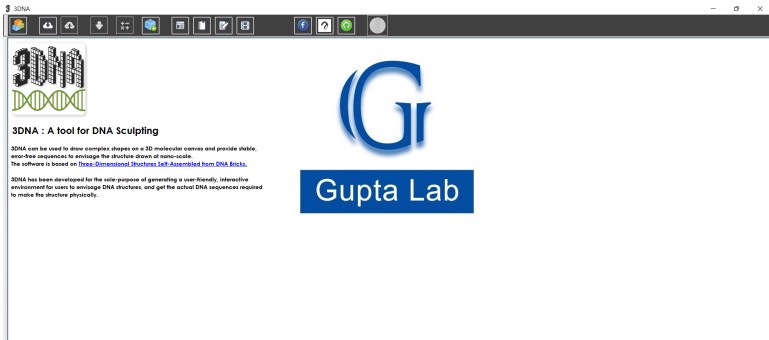

**Figure 3.** The introduction of the 3DNA Software GUI.

Some of the significant options are listed below:

(a) ⬜ option navigates the user to the 3D canvas.

(b) ⬇ option allows the user to save the output DNA sequence for visualization and eventual synthesis of DNA oligonucleotide sets.

(c) ➗ provides an estimation of the cost per DNA base in USD for a particular model.

(d) ⬜ navigates the user to the Advanced panel.

(e) 🧬 option allows user to visualize the DNA nanostructure.

A molecular canvas of any preferred dimensions can be sculpted by simultaneously removing individual or multiple voxels. For any given structure, the protector bricks (poly-1T DNA sequences to prevent unwanted hybridization) and boundary bricks (longer DNA sequences on the boundary surface) can be enabled or disabled to analyze or verify the stability of the structure experimentally. Larger structures made up of repeating small units called crystals can also be enabled in $X$, $Y$ and/or $Z$ direction(s). In the subsequent modules, the coordinates of the shape sculpted by the user are stored in voxels. To finish the sculpting process, these voxels are converted into DNA bricks and mapped to corresponding DNA sequences according to the procedure described earlier. This can be achieved by selecting a subset of sequences from a more extensive list to form the final shape. Alternatively, these DNA sequences can be generated directly using the random sequence generator or importing a predefined set of sequences. Upon finishing the modeling, three different visualization choices, namely full-canvas, planar, and elementary visualizations, are available to bring a nearly all-around rendering of the nanostructure. The software offers two formats to save the final DNA sequences on the local storage, .pdf and .csv, as the ultimate output.

Eventually, the user can access the whole data corresponding to the prescribed nanostructure, which includes the latest DNA sequences and the coordinates in a precise format in the output files. For better understanding, the workflow schematic in Figure 4 depicts the basic functionality of 3DNA.

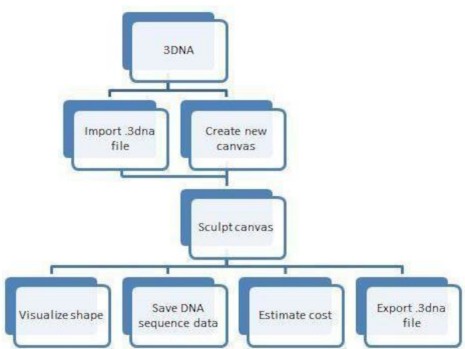

**Figure 4.** The flowchart for the detailed process of different modules in 3DNA.

Briefly illustrated features of 3DNA are described in the following subsections.

### 2.1. 3D Molecular Canvas

The 3D canvas, a cuboid figure, is the combined form of individual molecular 3D pixels, commonly known as voxels, where each voxel measures $2.5 \times 2.5 \times 2.7$ nm and represents an 8-nt duplex pattern. The shape is formed by the interlinkage of a domain with its respective complementary DNA strand, represented as a voxel. The collection of all such voxels is the entire 3D canvas. To avoid ambiguity, each face of the molecular canvas is differentiated by different colors. By default, green indicates the $XY$ plane, while grey and yellow indicate the $YZ$ plane and the $XZ$ plane, respectively.

Also, coordinates are determined by considering the top left corner of the starting face as the origin of the structure. This other terminology is set by identifying the starting block as the voxel associated with the origin with the coordinate $(0, 0, 1)$. One should note that the initial $Z$ axis plane, defined as the plane number 0, will be occupied by the protector bricks with poly-1T domains (composed of Thyamin bases) upon enabling the appropriate option, and hence the starting first $Z$ axis is identified as the plane number 1. To add more flexibility, enabling or disabling the protector bricks can be made independently.

### 2.2. Modeling/Sculpting Shapes

The user can specify the dimensions of the 3D canvas as input (Figure 5). It is noteworthy that the 3DNA 3.0 version also supports the expanded 13-base-pair voxel pattern [13] along with 8 nt bricks to build the DNA origami structure, as it is evident that 13 nt brick structures have improved melting temperature, and are relatively more stable as compared to 8 nt structures.

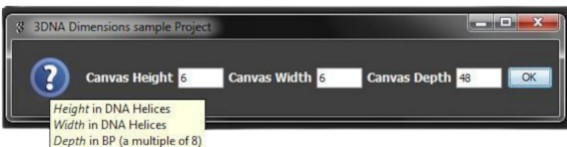

**Figure 5.** The canvas dimension input window.

To provide an intuitive experience, the interface allows the user to navigate through the blocks using the navigation panel placed on the left side of the window. The navigation panel comprises zooming, rotation, translation, and undo buttons for an improvised visualization in all three directions.

The input interaction with the interface can be done in two ways, mouse movements on the voxels individually or clicking the buttons in the left toolbar for further process. For example, we sculptured the 6 × 6 × 6 (48 bp) cuboid in Figure 6.

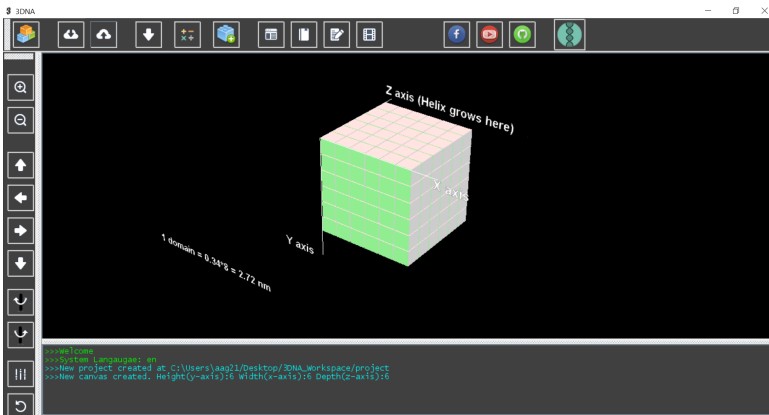

**Figure 6.** The front-end view of a 6 × 6 × 6 (48 bp) cuboid in the 3D modular canvas.

## 2.3. Advanced Panel

The GUI includes an exclusive Advanced Panel accessed from the taskbar. The panel (Figure 7) consists of functions ranging from remodeling the canvas to implementing constraints and features to the DNA sequences.

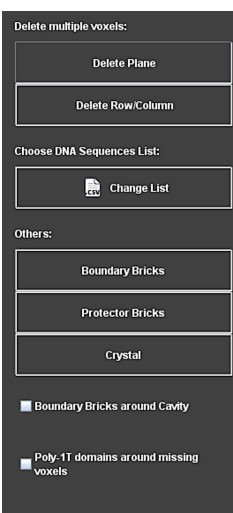

**Figure 7.** Advanced panel for added features.

In particular, a user can replace multiple components in the structure by choosing a random set of unique sequences offered by the interface. It is noteworthy that the default generated DNA sequences chosen from the list will be constrained sequences that automatically allow 40–60% GC content and Hamming distance up to 6 per domain, where domain length varies between 8-nt and 13-nt. Alternatively, the software allows users to import a fixed set of DNA sequences in .csv format for restructuring purposes. The interface accepts three different import options, such as 16-nt file or 32-nt file, or 48-nt file formatted in a predefined manner.

In the case of importing DNA sequences, one should keep in mind that for a preferred protector brick structure, the poly 1T domain strands should be contained in the imported file.

*2.4. Implementation*

Advanced functionality features on 3D structures, such as 48-nt Boundary bricks, Protector bricks, and Crystal formation, are present to enable or disable different forms of brick formation or alignments on the structure.

Specifically, concatenating a 16-nt boundary brick with a 32-nt brick along the *Z*-axis will result in a 48-nt boundary brick in different canvas planes. On the other hand, protector bricks can be added to the extremal corners along the *Z* axis. During the implementation, if users import their own set of DNA sequences, it may happen that a sequence cannot be integrated into the structure. In that case, a modification will be done by substituting the respective complementary domains of single-stranded DNA with poly 1T domains to proceed further upon choosing the appropriate option. Otherwise, the complementary strands remain in the sculpture. Moreover, analyzing the sculpture's stability and formation upon including or excluding a longer strand can be tested using the "Enable cavity boundary bricks" option.

As an exciting functionality, the interface enables the design of extended crystalline structures formed from repeating brick unites (Figure 7) expanded in all three dimensions and in different planes. To illustrate this, we considered the previous example of the $6 \times 6 \times 6$ (48 bp) cuboid and demonstrated the resultants in Figure 8.

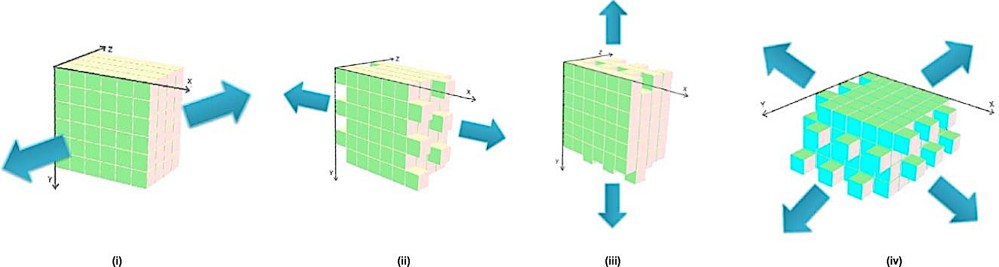

**Figure 8.** The crystal structure formation of the $6 \times 6 \times 6$ (48 bp) cuboid (Figure 6) along different directions. (**i**) elongated along *Z* axis, (**ii**) elongated along *X* axis, (**iii**) elongated along *Y* axis, and (**iv**) crystal formation in *XY* plane along *X* and *Y* axes.

*2.5. Visualization*

The visualization module is added to the interface to overcome the constraint of graphical rendering and to provide a more comprehensive insight into the structural mechanism of the sculpture. To envisage the DNA nanostructure, the module offers three kinds of visualizations (Figure 9), namely, Full Canvas, Elementary, and Plane.

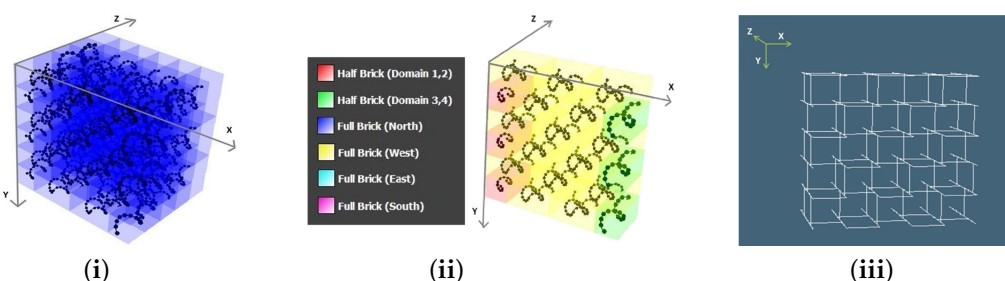

**Figure 9.** The visualization canvas of the $6 \times 6 \times 6$ (48 bp) cuboid (Figure 6) provides three different rendering modules. (**i**) Full canvas visualization, (**ii**) Plane visualization with color differentiation markers, and (**iii**) Elementary visualization.

The full canvas visualization provides the helical interpretation of the single-stranded DNA sequences that interacted in a LEGO®-like manner. The plane visualization is the brick-based color-encoded helical representation of the structure in a specific plane. The full canvas and plane visualizations of the 6 H $\times$ 6 H $\times$ 48 BP cuboid (Figure 6) are depicted in Figure 9i,ii, respectively. It should be noted that the full canvas visualization depends on

the specification of CPU working capability. Hence, as a lightweight alternative, 3DNA enables the "Individual plane visualization" option to render higher dimensional canvases with dimensions exceeding 6 H × 6 H × 48 BP.

In a different approach, the elementary canvas visualization is focused on the crystal-like formation of the structure. The DNA sequences are represented in arrow-like figures to render the sculpture at the nanoscale. Figure 9iii illustrates the elementary visualization.

The following category summarises the flexibility of the 3DNA software by considering a handful of options that can significantly enrich the usability of the software.

### 2.6. Silent Features

- Import and Export: To achieve more adaptability, the option of exporting a current project is available to the users. The current project will be stored as a .3dna file by availing the Export button. This feature also enables importing an existing project exported previously with the .3dna extension, which can be opened and modified.
- Importing user's DNA sequence: The user also has the ability to import a set of DNA sequences in .csv format to create shapes. On a brief technical note, the "Import sequence module" can take 3 .csv files to complete the molecular canvas, each one for half bricks (16 nt .csv file), full bricks (32 nt .csv file), and boundary bricks (48 nt .csv file) in specific formats.
- Structure filtration for stability: A developed module is attached that avoids DNA structures, allowing improper folding to test the topological connectivity for stability using the Graph Theory approach.
- Output analysis: Further statistical analysis on the output set of sequences can be done using the "Graphical Analysis option", which can be useful for conducting laboratory-based experiments. It also introduces an abrupt idea about the structural tolerance and stability of a self-assembly DNA nanostructure containing indistinguishable bases in domain sequences. The option represents the frequency of pairs of 8-base domains classified by the factor of containing 8, 7, or 6 identical bases among the sample space of the 432 domains. Figure 10 shows the analysis of the 6 H × 6 H × 48 BP cuboid (Figure 6) in 432 domains.
- Cost estimator: 3DNA also has an inbuilt estimator function that evaluates the experimental cost in USD by considering the number of nucleotide bases used in the process.
- While generating the structure, the user can undo the last step using the UNDO button available in the software.

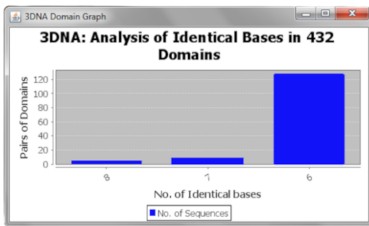

**Figure 10.** The Bar Diagram analysis of the 6 H × 6 H × 48 BP cuboid in 432 domains [50].

### 3. Conclusions

This work introduces and demonstrates the 3DNA software package, which can be used to design complex nanostructures based on the DNA brick nano-fabrication method using either 8-base pair or 13-base pair voxel patterns. The software provides an easy-to-use 3D molecular canvas to design different types of shapes and extended topologies, along with modules for the analysis of topological connectivity and corresponding compositional control over oligonucleotide sequences, using randomly chosen unique DNA sequences that allow 40–60% GC-content to favor thermal stability of the structure. In addition, the user can also utilize 3D rendering to visualize the designed nanostructure. As added features, the 3DNA GUI also accepts DNA sequences from pre-set structures provided

by the user and performs cost estimation based on inputted prices for oligonucleotide synthesis for practical usage. For the future scope, the hexagonal brick structure feature to design DNA nanotubes used for biomedical studies [51–53] will be added to the 3DNA software to make it more functional.

While the patterning of carbon nanotubes [36] or the implementation of molecular circuits [35] onto the surfaces of DNA brick structures are so far the most promising studies that only begin to scratch the surface of the potential for this technique, we envision that its wider availability will benefit a variety of approaches where functionality scales with the availability of a functional surface area. In these cases, particularly for recently-introduced strategies where the surfaces of DNA origami structures are used as coding substrates for cryptography [54], or data storage [55], the possibility to arbitrarily extend the surface area using larger sets of bricks or crystallization approaches could provide an expansion of the technique beyond a basic proof-of-concept.

It should be worth mentioning that the design of a topologically interconnected and consistent set of oligonucleotides using our software or any CAD package for generating complex DNA nanostructures is no guarantee that the final product will assemble to its intended shape. Even for the popular method of scaffolded DNA origami, topologically flawless designs made with the standard software Cadnano can nevertheless lead to misfolded structures due to kinetic traps, internal strain, or other factors [38]. A priori modeling and visualization of structures with analysis tools can help to detect such issues concurrent to the design process, however, these are not foolproof solutions. For the DNA Brick method, there are several structural and topological features that are not present in the more commonly used method of DNA origami, most notably the lack of a scaffold strand. This can cause structures to be more sensitive to certain types of design motifs which, while topologically interconnected, are nevertheless locally instable due to the lack of a stabilizing scaffold strand running throughout the structure. Indeed, the insertion of a short "seeding strand" into brick structures has been shown to have a positive impact on assembly [27]. While for DNA origami, there was at least one early study that examined the impact of certain design features such as staple-break positioning on the assembly of the final product [56], such systematic studies have not been carried out with DNA bricks, adding some degree of inherent risk to using the technique. Ultimately such detailed and systematic studies to empirically test the impact of specific design motifs on structural assembly and overall stability can be aided by CAD software, and also provide valuable parameters for iterative improvements of the software itself.

Lastly, although this software exhibits significant practical utility, the imperative of carrying out experimental validation cannot be overlooked. Thus, to bridge the divide between theoretical concepts and real-world applications, future endeavors will focus on presenting empirical evidence through rigorous experimental generation and visualization of DNA brick structures, underscoring its pertinence and capacity to address practical issues.

**Author Contributions:** S.K.G., F.J. and A.A. coded the basic software and extended modules. M.K.G. conceptualized the software package. M.S. and D.M.S. supported module integration, design of overall software GUI, and conceptualization of extended modules. A.A., S.D., D.L., D.M.S. and M.K.G. wrote and edited the manuscript. All authors have read and agreed to the published version of the manuscript.

**Funding:** David M. Smith and Manish K. Gupta acknowledge support from Deutsche Akademische Austauschdienst (DAAD) (PPP 57453411 "VecDNAStor") and Ministerium für Wirtschaft, Arbeit und Wohnungsbau Baden-Württemberg (231201553 AsphyxDX). Manish K. Gupta was supported by the Department of Science and Technology (DST/INT/DAAD/P-14/2019), Govt. of India.

**Data Availability Statement:** 3DNA software can be downloaded from http://guptalab.org/3dna/ (accessed on 15 December 2023). The web page also provides the user manual and supplementary material. Additionally, the source code is available at https://github.com/guptalab?tab=repositories (accessed on 15 December 2023). Any software-related query can be addressed at 3dna@guptalab.org

(accessed on 15 December 2023). The updated 3DNA 3.0 is tested on the Java JDK version 1.6.0_45, enabled with Java3D (version 1.5.1). For the 3DNA 3.0 demo illustration, the instructional video is available https://www.youtube.com/watch?v=0XNlnEVsbR4&ab_channel=ManishGupta (accessed on 15 December 2023).

**Acknowledgments:** The authors acknowledge the help from Bruno Lowagie (https://itextpdf.com/, (accessed on 12 December 2023)), whose free open-source libraries have been used for generating bar codes and PDFs. The authors are thankful to Param Parekh and Paavan Parekh for their assistance in creating the supplementary resources (documentation and tutorial videos) for the 3DNA 3.0 version. All authors have read and agreed to the published version of the manuscript.

**Conflicts of Interest:** The authors declare no conflict of interest.

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
