# Peer review of "3DNA: A Tool for Sculpting Brick-Based DNA Nanostructuresâ€"

_2674-0583, doi:10.3390/synbio1030016_

Round 1
Reviewer 1 Report
Comments and Suggestions for Authors
DNA especially its double-stranded form since 1953 has been at the centre of different types of scientists. For many years the structure of the double helix was obscured. Due to their flexibility and possibility toward higher structure formation the prediction of the object is still attractive. As the authors mention in their article the aptamer strategy (therapeutic), DNAzymes, and origami DNA are just the few that scientists undercover. On the other side, the growth of the nanotechnology and corresponding therapeutic strategy required a simple, fast method/software/algorithm that will give reasonable results. In the article, the authors present the new “software” to predict the 3D structure of nanomaterial-based/constructed on DNA. As it was with different softwares the time will show its usefulness. In the article, I did not find points to meritocratic critics. However, authors should prepare the figures in higher resolution and mention the previous work of Olson, Lu and Westhof (standard reference frame). In conclusion, I can recommend this article for publication.
Author Response
We appreciate the referee’s positive assessment of our manuscript and recognition of the potential broad applicability of the software we provide. It is indeed our hope that, over time, 3DNA will prove to be a useful addition to the number of CAD software packages for DNA nano-engineering, and will ultimately increase the overall usage of the revolutionary brick method. As per their request, we have integrated the following revisions:
The referee is entirely correct that the earlier visualization/analysis software of the same name from Lu and Olsen (recently updated to the name w3DNA for its web version) should be mentioned within the manuscript, at the very least to avoid confusion amongst readers. While the earlier and recently updated software from Lu and Olsen is strictly for the purpose of visualizing and analyzing relatively simple DNA/RNA motifs based on protein database (PDB) files, it is nevertheless our opinion that this type of program forms a convenient, if not necessary complement to CAD software such as we present in our manuscript. Therefore, we have added an additional paragraph to the introduction (line 104-113) generally dealing with this type of tool, with special emphasis to the original Lu and Olsen work, along with another similar program MFold, which was initially released in a similar timeframe.
Two further sentences (line 131-136) are added on the same page to explicitly differentiate the intended function of our current CAD program from the earlier modeling software of the same name.
Additionally, the figures have all been updated to be of a higher quality and resolution. We hope that this current quality is sufficient for publication.
Reviewer 2 Report
Comments and Suggestions for Authors
In this manuscript authors claimed to have developed a CAD tool for designing DNA bricks. While the software looks fine, for the scope of SynBio, authors need to show the sequences generated can actually assemble into the expected shape. Either AFM or TEM images will be expected. Therefore, a major revision is recommended for authors to prove the designing capability of their software.
Comments on the Quality of English LanguageModerate editing of English language required
Author Response
We appreciate the referee’s overall positive assessment of the software and do agree that an experimental demonstration of assembled structures, directly visualized by AFM or TEM would be the final verification of its success in outputting sets of self-assembling oligonucleotide sequences. This approach was taken, for example, in the first CAD program for the DNA origami method, Cadnano (Douglas et al., 2009, doi:10.1093/nar/gkp436). In the present study, within the requested time constraints for addressing referee comments, as well as in the practical technical and financial constraints of the main participating groups, this is unfortunately not a feasible option. Based on practical knowledge of previously experimental work, the entire process to design several structures, order the hundreds of constituent oligonucleotide strands, carry out and optimize assembly (in addition to any necessary purification protocols), and obtain high-quality images is an endeavor that would require several months of work, and the necessary addition of a collaborator with capacities to record high-quality images.
While we are confident in 3DNA’s generation of correct sets of DNA sequences according to the topological structure defined by the user, the referee’s comment does bring up an equally important point which we will address in the text. While the handful of published studies reporting or using the DNA brick method do show the successful instances of the method, there are nevertheless many structural and topological challenges that are not present in the more commonly used method of DNA origami. These can make structures more sensitive to certain types of motifs which, while topologically robust, are nevertheless not practically stable due to the lack of a single scaffold strand running throughout the structure. In short, there is no guarantee that a topologically consistent and interconnected structure will assemble properly. While for DNA origami, there was at least one early study which examined the impact of certain design features such as staple-break positioning on assembly of the final product (Ke et al., 2012, 10.1039/C2SC20446K), such systematic studies have not been carried out with DNA bricks, adding some degree of natural risk to using the technique. Therefore, we have added a new paragraph at the end of the manuscript (lines 317-337) to point out this important aspect, and propose it as a general way forward to motivate scientists in the field.
Reviewer 3 Report
Comments and Suggestions for Authors
The manuscript presents a software to help in design DNA structure, and it would be very important to show the geometry difference in addition to a cubic one, such as tetrahedral pyramid which is commonly used for bio sensings, and capsules in drug delivery, and etc. It would be better if this software can demonstrate strength in geometry designs.
Comments on the Quality of English Languageno
Author Response
We appreciate the suggestions from the referee about further types of DNA architectures. Indeed, the small tetrahedral structure, first introduced by Goodman et al. in 2005 (doi: 10.1126/science.1120367) is one of the most used DNA nanostructures in a wide variety of applications. Nevertheless, the software we present here is specifically meant to be a tool for the DNA brick method, which does have certain inherent geometric limitations in the types of structure it can generate. Specifically, the DNA brick method is not suitable for the small wireframe DNA structures formed from only a small number of oligonucleotides, or wireframe structures in general, since it is geometrically constrained to a voxel-like grid, where structures are effectively conceptualized in a block-like fashion. Instead, several CAD programs already exist for creating this type of structure, and also far more complex wireframe DNA nanostructures. These are largely covered in an earlier review paper we authored (Glaser et al., 2021, doi:10.3390/molecules26082287). The program we introduce here is intended to fill the gap for the brick method, since up to now, no dedicated CAD software exists yet for this design strategy.
Round 2
Reviewer 2 Report
Comments and Suggestions for Authors
Thank you for the revision from authors. While it would be ideal to include experimental evidence to prove the output the new CAD software, it could be accepted as a theoretical paper as long as authors stated clearly about the importance of experimental proof.
Comments on the Quality of English LanguageModerate editing of English language required
Author Response
Dear Editors and Referees,
We are grateful for the positive feedback and final suggestions to improve our manuscript. We also appreciate and agree with the referee’s assessment that the work is suitable to serve a standalone theoretical work, even though experimental verification through high-resolution imaging of several structures would be ideal. We have taken their suggestion to clearly note the purely theoretical nature of this study, as well as point out the future necessity of further experimental work.
In order to emphasize this point, we have made the following changes to the manuscript:
In Line 10 (abstract), we removed the word “experimentally” from the sentence previously stating “3DNA is developed to provide a streamlined, real-time workflow for designing and experimentally implementing this type of 3D nanostructure by integrating different visualization and design modules.” We feel this will eliminate any unintended impression to readers focusing only on the abstract that the manuscript is also reporting experimental verification.
Near the end of the introductory section, we add the sentence “While a rigorous experimental demonstration of a variety of different designs created using this software should still be carried out, the numerous beneficial features of 3DNA validate the theoretical underpinnings, demonstrating its relevance and potential to solve practical problems, ultimately enhancing its value and impact.” This is intended to draw attention to the eventual need for experimental verification.
We have added a new closing paragraph of the manuscript which states “Lastly, although this software exhibits significant practical utility, the imperative of carrying out experimental validation cannot be overlooked. Thus, to bridge the divide between theoretical concepts and real-world applications, future endeavors will focus on presenting empirical evidence through rigorous experimental generation and visualization of DNA brick structures, underscoring its pertinence and capacity to address practical issues.”
Furthermore, the referee indicated that some improvement could be made to the English usage in the manuscript. Therefore, a native English speaker has carefully read through the manuscript and made several minor changes to remove awkward wordings, punctuation errors or other mistakes in the language. These are marked within the manuscript.